# The impact of "male clinics" on health-seeking behaviors of adult men in rural Kenya

Justine Dowden[1¤a], Ivy Mushamiri[2], Eric McFeely[1¤b], Donald Apat[3], Jilian Sacks[1¤c], Yanis Ben Amor[1] *

**1** Center for Sustainable Development, Earth Institute, Columbia University, New York, NY, United States of America, **2** Department of Epidemiology, Mailman School of Public Health, Columbia University, New York, NY, United States of America, **3** Columbia Global Centers East and Southern Africa, Nairobi, Kenya

¤a Current address: Center for International Earth Science Information Network | Earth Institute, Columbia University, New York, NY, United States of America
¤b Current address: Kaiser Permanente Northern California | Oakland, Oakland, CA, United States of America
¤c Current address: Clinton Health Access Initiative, Boston, MA, United States of America
* yba2101@columbia.edu

## Abstract

### Background

In most parts of the world, men access health services less frequently than women, and this trend is unrelated to differences in need for services. While male involvement in healthcare as partners or fathers has been extensively studied, less is known about the health-seeking behavior of men as clients themselves. This interventional research study aimed to determine how the introduction of male-friendly clinics impacted male care-seeking behavior and to describe the reasons for accessing services among men in rural Kenya.

### Methods and findings

We questioned men to assess utilization and perceptions of existing health clinics, then designed and evaluated a "male clinics" intervention where dedicated male health workers were hired for one year to offer routine, free services exclusively to men within existing healthcare facilities. Results were compared between data from Male Clinics in specific health facilities, the same facilities concurrently, nearby control facilities concurrently, and intervention facilities historically.

Costs of services, distance to facilities, and quality of care were the main barriers to healthcare access reported. The number of total visits was significantly higher than control groups (p<0·0001). In the intervention group, 18·6% of visits were for a checkup compared to almost none in control groups. The most common diagnoses overall were upper respiratory tract infections, malaria and injury. A major limitation of this study is the non-comparability in information captured using the Male Clinic registers compared to control registers.

### Conclusions

Costs and quality of services deter men from seeking healthcare. The introduction of male-friendly health services could encourage men to seek preventive care and increase service uptake.

**Data Availability Statement:** The anonymized data supporting the study is available within the paper and its Supporting Information files.

**Funding:** Research funded by UNAIDS (https://www.unaids.org/en) under grants PG005551

(received by YBA) and P005210 (received by YBA). The funders had no role in study design, data collection and analysis, decision to publish, or preparation of the manuscript. For the research reported in this publication, IM was supported by the National Institute of Allergy and Infectious Diseases of the National Institutes of Health under Award Number T32AI114398. The content is solely the responsibility of the authors and does not necessarily represent the official views of the National Institutes of Health.

**Competing interests:** The authors have declared that no competing interests exist.

## Introduction

Access to free services remains a significant challenge to receiving quality healthcare in resource-limited settings and is a cause of underutilization of healthcare by men [1]. While male involvement in healthcare as partners or fathers has been extensively studied in the developing world context [2–6], far less emphasis has been placed on the health-seeking behavior of men as clients themselves. Often, studies analyzing men's attendance at rural health clinics in low-income settings have focused on their involvement in programs related to maternal and child health, especially programs to decrease mother-to-child transmission of HIV [5, 7–10]. However, far fewer programs and research initiatives in low-income countries focus on men exclusively as independent agents seeking access to healthcare [11]. This is problematic because seeking care is often seen as counter-normative for men, particularly in patriarchal societies, and is instead an activity viewed as particularly necessary for women and children [12–14]. Furthermore, clinics are often viewed as "female spaces," because females usually make up most of the patient and caregiver population in these settings [12]. As a result, it is common for men to only seek care during emergencies or in the later stages of preventable illnesses [15–17].

The "male clinics" intervention aimed to narrow this gap in male health-seeking behavior in western Kenya, where services are primarily accessed by women and children. Since services are also mostly provided by female health workers, the research hypothesis of the qualitative study was that men's concerns over confidentiality and views about gender norms are barriers to their uptake of services at clinics. Preliminary research was conducted to learn the reasons local men were not more frequently visiting clinics, and subsequently a "male clinics" (MCs) intervention was implemented to address the identified challenges. In viewing men as patients in their own right and providing a space to meet their individual needs confidentially and sensitively, we sought to create a more enabling clinical environment with the goal of ultimately increasing men's uptake of health services. The paper highlights the benefit of a general health approach, rather than a disease-based approach, to engage men in health services.

By appointing male health workers and directing services to men alone in a separate area within a clinic, we aimed to create an incentive for men to seek care. The hypothesis was that a male-friendly space would improve men's health-seeking behavior and increase uptake of care by reducing cultural barriers, particularly in the case of stigmatized or sensitive conditions. Also, targeted services were intended to improve efficiency as an added incentive, since patients are often required to wait a long time for care at clinics in resource-limited settings.

## Materials and methods

### Study area

The study site was located in Sauri, Siaya county in western Kenya within a Millennium Villages Project (MVP) site. The MVP context has been described previously [18–19]. Sauri contains 11 villages and the total catchment population for the health facilities is 73,089. Adult men over 18 years of age make up approximately 13·3% (n = 9,713) of the total population [20]. There are 10 health facilities, one of which offers free care for all services to men. In other clinics, care is only completely free for pregnant women and children under five; some basic care is free for all. Only Ramula and Nyawara intervention clinics offered free voluntary medical male circumcision, while the others did not. Both clinics were low volume sites.

### Preliminary focus groups and trial

In the first phase of this study (October-December 2013), a qualitative questionnaire was fielded with 124 men over 18 years old in Sauri district to assess utilization and perceptions of

local clinics. MVP staff led recruitment. Community Health Workers (CHWs) spread awareness of the questionnaire during home visits and others recruited by word-of-mouth during community events such as "men's health days." A convenience sample was established by selecting men from communities with differing proximities to clinics.

Kenyan data clerks who worked for MVP were trained in survey administration. The questionnaire was available in English, Swahili and Luo. The interviewers conducted structured focus group discussions and individual interviews, including questions about frequency of and reasons for past clinic visits, along with open-ended questions about reasons men would visit a clinic, obstacles preventing men from accessing care and possible improvements to the health centers.

Following the focus groups, MVP conducted a one-month pilot MC at two clinics in Sauri. As part of this pilot, we organized a "men's health day," which was a sensitization day involving a male nurse at each clinic providing education to male visitors primarily about sexual health, as per focus group recommendations. Men were then seen by a male clinical officer hired by MVP for individual consultation at a unit separate from the clinic that caters to women and children; services were free.

## Establishment of male clinics

Following feedback from the one-month pilot, we designed an MC intervention, hiring two dedicated male health workers to be stationed in a designated area within existing healthcare facilities to offer free services exclusively to male patients on a specific day of every month. We carried out this intervention for 12 months at five local health facilities to determine the impact of a male-friendly clinic on male attendance rates at the health centers, and its impact on specific health-seeking behaviors associated with free simplified access.

## Intervention activities

The two male health workers were stationed for several daytime hours once per month at five clinics—Lihanda, Marenyo, Nyawara, Ramula, and Sauri—in an isolated area within each clinic to ensure privacy. During consultations, the health workers filled out a register with basic information about the visit, but no identifiable information was recorded to comply with ethical review board requirements (S1 Appendix).

The health workers at the MC offered a range of services for common ailments (Table 1). Preventive care was also available, including HIV testing, STI screening, and blood pressure checks, and men were referred if necessary.

## Evaluation design

Patient registers were collected from clinics in four groups: 1) "Intervention," which was data collected on MC days, 2) "Concurrent control at intervention clinics," which was data from the same clinics where the intervention took place during the same time period, but on days when the MC did not occur, 3) "Concurrent control at different clinics," was data from five clinics which did not have MCs during the same time period as the intervention, and 4) "Historic control," which was data from intervention clinics two years prior from November 2012 to November 2013.

MCs were open during the same hours as control clinics.

Because no identifiable information was collected, the unit of analysis was visits, not people, with the exception of the age variable. Median age was calculated by excluding the subsequent visits of those who had visited MCs more than once to avoid double counting.

**Table 1. Common ailments that Male Clinics were equipped to treat.**

| |
|---|
| Malaria |
| URTI |
| STI |
| Enteric fever |
| Neuritis |
| Gastroenteritis |
| Fungal infections |
| Hypertension |
| Diabetes |
| Epilepsy |
| Arthritis |
| Allergy |
| Injury/trauma |
| Sexual dysfunction |
| UTI |
| Peptic ulcer disease |
| Pneumonia, asthma, pleurisy/LRTI |
| Dental problems (referral) |
| Myalgia |
| Cellulitis |
| Other |

## Outcomes of interest and measurement

Outcomes of interest were divided into five groups: reason for visit, service utilization, type of diagnosis, quality of care, and the "male-friendliness" of the MCs. We aimed to understand how many visits were made by sick patients compared to healthy patients coming for check-ups, whether uptake of services increased when an MC was offered, for which ailments men sought care, whether MC clinicians appropriately diagnosed patients, and whether patients were visiting MCs for sensitive reproductive health issues.

The reason for visit was recorded as either "check-up" or "sick," as per MC clinician assessment, and was compared to the control groups. The MC clinician also recorded whether each visit was a first or follow up visit.

To measure service utilization, we calculated the number of visits on a monthly basis as a proportion of the total male population in the catchment area and compared the number of visits at MCs with the number of visits in all control groups. We also calculated the proportion of total visits that occurred monthly.

Quality of care was measured by assessing how frequently the MC clinician provided proper treatment for six common diagnoses that corresponded with standardized treatments (Table 2). We chose these ailments because they had a finite number of treatment options in this particular setting.

Also, the dates of study collection among the concurrent control at different clinics group was limited to July-November 2015, while data was collected from November 2012-November 2013 in the historic control group and November 2014-November 2015 in the remaining two groups.

## Statistical analysis

Statistical analyses were conducted using SAS 9·4, STATA 14·1, SPSS 22·0, and MS Excel 2016 software. The preliminary questionnaire was analyzed using SPSS after combining open-

**Table 2. Common diagnoses at Male Clinics and corresponding standardized treatments.**

| Diagnosis | Treatment |
|---|---|
| Diabetes | Metformin |
| Hypertension | Hydrochlorothiazide |
| Epilepsy | Anti-epileptics[a] |
| Fungal infection | Anti-fungals[b] |
| Peptic Ulcer Disease (PUD) | Omeprazole |
| Malaria | Artemether/Lumefantrine |

[a] Includes phenobarbital, diazepam, tegretol

[b] Includes clozole, griseofulvin, gentian violet

ended responses into similar categories. To test for a difference in the number of visits made as a proportion of the total male population in the catchment area by group as a measure of service utilization, a two-sample t-test was conducted for each intervention-control group comparison and an ANOVA test was used to compare across groups. Bonferonni adjustments were made to account for multiple comparisons (alpha/3). To assess correct treatment for diagnosis by group as a measure of quality of care, a chi-square analysis was conducted comparing the proportion of diagnoses that were correctly treated by intervention or control group. Quality of care was measured by calculating the proportion of correct treatments for each diagnosis in the intervention group and was assessed using chi-square tests with a Bonferonni adjustment to account for multiple comparisons. A chi-square test was also used to assess the proportion of visits that were check-ups, and the proportion of times a provider-initiated testing and counseling (PITC) was conducted among patients who did or did not know their HIV status.

Finally, we investigated how many men were diagnosed with STIs and/or sexual dysfunction at MCs compared to the control groups, with the expectation that men were more likely to want to address these sensitive issues in an MC environment versus a regular clinic. A chi-squared test was used to compare the proportion of diagnoses that were STIs and sexual dysfunction between the intervention group and all control groups.

## Results

### Preliminary questionnaire findings

Cost of services, distance, and cost of travel were the most commonly reported barriers to visiting clinics among men who had never visited a facility (n = 112). Also, only 52·9% (n = 63) of men overall reported satisfaction with healthcare providers. Common suggestions for improvements to providers included being more respectful (32·7%, n = 33) and reducing wait times (23·8%, n = 24). Participants' most common fears about visiting facilities included a lack of confidentiality (35%, n = 7), time delays and inefficiency (20%, n = 4), and concerns about HIV testing (10%, n = 2) (Table 3). Additionally, 86·3% (n = 107) of respondents felt that women were most likely to use health facilities; just 2·4% (n = 3) said adult men were most likely to do the same. This feedback informed the design of MCs.

### Male clinics findings

#### Demographics

The median age of MC patients was 32. Men who visited the MC multiple times were included in this calculation only for their first visit; ages of patients coming for subsequent visits were

**Table 3. Select preliminary questionnaire findings.**

| | Total number of responses | % |
|---|---|---|
| **Satisfaction with health providers** | 119 | |
| Satisfied with providers' competency | 65 | 54.6% |
| Satisfied overall | 63 | 52.9% |
| Satisfied with provider's attitudes | 55 | 46.2% |
| **Believe improvements could be made to providers** | 116 | 93.5% |
| Yes | 104 | 89.7% |
| No | 12 | 10.3% |
| **Suggestions for improvement to providers** | 135 | |
| More respectful | 40 | 29.6% |
| More timely | 30 | 22.2% |
| More staff | 25 | 18.5% |
| Staff provide medication | 15 | 11.1% |
| Better trained | 13 | 9.6% |
| Staff provide better urgent care | 4 | 3.0% |
| Ward for men | 2 | 1.5% |
| Staff provide free care | 2 | 1.5% |
| Improved hygiene | 1 | 0.7% |
| More male providers | 1 | 0.7% |
| Less corrupt | 1 | 0.7% |
| Mortuary | 1 | 0.7% |
| **Common reasons men visit health facilities** | 121 | |
| Treatment during illness (usually severe) | 80 | 66.1% |
| HIV services | 26 | 21.5% |
| Malaria | 6 | 5.0% |
| Health information/general check-ups | 3 | 2.5% |
| Circumcision | 3 | 2.5% |
| Accompanying others | 2 | 1.7% |
| First aid | 2 | 1.7% |
| STI services | 1 | 0.8% |
| Overall health | 1 | 0.8% |
| Obstetrics | 0 | 0.0% |
| Family planning | 0 | 0.0% |
| **Common fears about seeking care among men with fears** | 25 | |
| Not confidential | 7 | 28.0% |
| Delays / inefficiency | 4 | 16.0% |
| Scared of HIV test/result | 2 | 8.0% |
| Fear of blood draws / injections / taking medication | 2 | 8.0% |
| Service will be inaccurate / incorrect | 2 | 8.0% |
| Harshness of how staff treat patients | 1 | 4.0% |
| Expense | 2 | 8.0% |
| **Largest barrier to visiting health facility among men who have never visited a facility** | 112 | |
| Cost of services | 74 | 66.1% |
| Distance | 15 | 13.4% |
| No time | 12 | 10.7% |
| Cost of transport | 4 | 3.6% |
| No barriers | 3 | 2.7% |

*(Continued)*

**Table 3.** (Continued)

|  | Total number of responses | % |
|---|---|---|
| Not helpful | 1 | 0.9% |
| Fear of HIV test | 1 | 0.9% |
| Attitude of health staff | 1 | 0.9% |
| Majority of staff women | 1 | 0.9% |

excluded. The ages ranged from 15 to 80-years-old. Many men were farmers as agriculture is the main economic activity in the region. Most likely the younger patients were often students at a nearby university.

## Number of visits

Comparison by ANOVA revealed that the number of visits to MCs as a proportion of the total catchment population was higher in the intervention group than the number of visits to a clinic (by population) made by men in any control group ($p < .0001$).

The mean visits (and standard deviation) for each group were 3·77 (1·70) for the concurrent control at different clinics group, 21·26 (15·89) for the historic control group, 26·37 (19·32) for the intervention group, and 20·15 (15·35) for the concurrent control at intervention clinics group (Fig 1). Additionally, out of 571 MC visits, half (50·1%, n = 286) were re-visits (Table 4).

## Reasons for visits

The reason for visit was recorded as either "checkup" or "sick." In all control groups combined, there were just 34 checkups recorded, 1·2% of total visits in those group, and zero in the

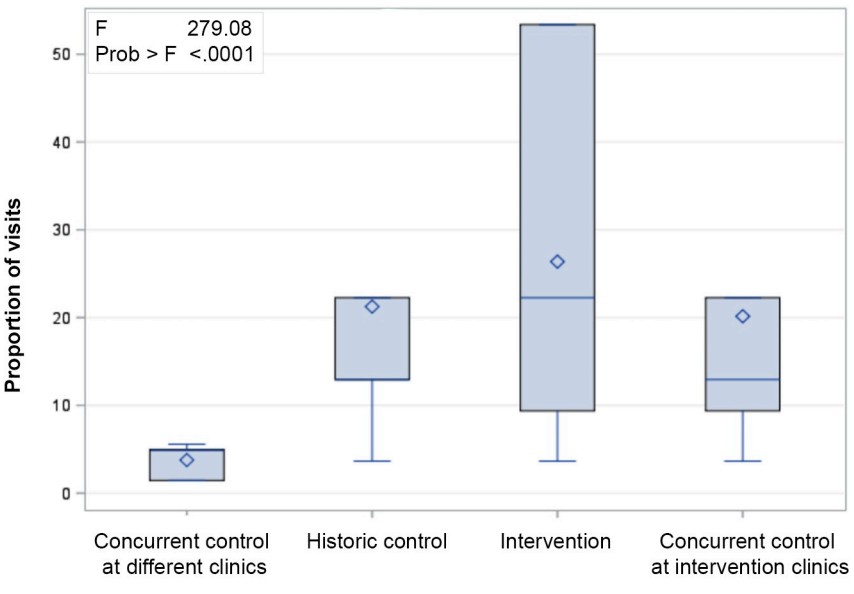

**Fig 1. Monthly visits to clinics as a proportion of total male population in catchment area by intervention group (ANOVA).**

**Table 4. Proportion of repeat visits to Male Clinics.**

| Clinic | Total visits to intervention clinics | Amount of re-visits to Male Clinics | Re-visits to Male Clinics as a proportion of total visits |
|---|---|---|---|
| Lihanda | 182 | 95 | 52.2% |
| Marenyo | 107 | 52 | 48.6% |
| Nyawara | 67 | 20 | 29.9% |
| Ramula | 83 | 33 | 39.8% |
| Bar Sauri | 132 | 86 | 65.2% |
| Total | 571 | 286 | 50.1% |

concurrent control at different facilities group. Conversely, 20·2% (n = 115) of visits in the intervention group were for checkups instead of in response to a health problem.

## Most frequent diagnoses

The most common diagnoses across all groups were upper respiratory tract infections (URTI), malaria and injury. The intervention group had the lowest proportion of URTI diagnoses (20·6%, n = 136). On average, 26·2% (n = 1066) of visits resulted in a URTI diagnosis in all groups. For malaria, the proportion was lowest (20·6%, n = 136) in the intervention group. Injuries were diagnosed at 9·8% (n = 65) of MC visits (Table 5). During checkups in the intervention group (n = 115), the top two diagnoses were URTI (15·7%, n = 18) and hypertension (12·2%, n = 16).

## Reproductive health services

We investigated how many men were diagnosed with STIs and/or sexual dysfunction at MCs compared to the control groups in pursuit of the "male-friendly" hypothesis that men were more likely to want to address these sensitive issues in an MC environment versus a regular clinic. We compared the number of STI diagnoses in control groups with the intervention group. STI diagnoses were the result of 2·6% of MC visits (n = 14), 3·6% of visits in the "concurrent control at different facilities" group, 3·3% in the historic control group, 2·6% in the "concurrent control at intervention facilities" group, and 2·97% (n = 90) of visits in all control groups combined. Differences in STI diagnoses across all groups were not statistically significant (p = 0·5043).

Sexual dysfunction was a recorded diagnosis at two visits in the intervention group and at zero visits in all control groups. The small number is not enough to draw a meaningful conclusion.

Within the intervention group, 25 syphilis screenings occurred, of which five were positive. After malaria, PITC for HIV was the second most common test at all intervention clinics except for Ramula, where it was the most common. PITC was offered 184 times and made up 27·4% of all tests (n = 672). There was no statistically significant difference in PITC offering by clinic (p = 0·11).

## Quality of care

Rates of correct treatment for diagnosis by group varied significantly. The intervention group had the smallest proportion of correct treatments by diagnosis (82·9%, n = 165) compared to all control groups (p<0·0001). This figure did not vary greatly among control groups, ranging from 93·3% in the historic control group to 90·9% in the "concurrent control at different clinics" group. Out of six common diagnoses, malaria and epilepsy were treated correctly 100% and 95% of the time respectively in the intervention group while fungal infections were treated

**Table 5. Frequency of diagnoses in all groups.**

| Diagnosis | Intervention group | | Concurrent control at different facilities group | | Historic control group | | Concurrent control at intervention facilities group | |
|---|---|---|---|---|---|---|---|---|
| | n | % | n | % | n | % | n | % |
| Allergy | 20 | 3.0% | 5 | 0.7% | 12 | 1.5% | 12 | 0.6% |
| Arthritis | 14 | 2.1% | 9 | 1.3% | 8 | 1.0% | 14 | 0.7% |
| Cellulitis | 0 | 0.0% | 9 | 1.3% | 11 | 1.4% | 39 | 2.1% |
| Dental | 7 | 1.1% | 29 | 4.1% | 6 | 0.7% | 9 | 0.5% |
| Diabetes | 11 | 1.7% | 0 | 0.0% | 4 | 0.5% | 13 | 0.7% |
| Enteric Fever | 23 | 3.5% | 16 | 2.2% | 5 | 0.6% | 2 | 0.1% |
| Epilepsy | 10 | 1.5% | 4 | 0.6% | 9 | 1.1% | 23 | 1.2% |
| Fungal infection | 17 | 2.6% | 12 | 1.7% | 17 | 2.1% | 13 | 0.7% |
| Gastroenteritis | 27 | 4.1% | 26 | 3.6% | 22 | 2.7% | 63 | 3.4% |
| Hypertension | 25 | 3.8% | 12 | 1.7% | 5 | 0.6% | 26 | 1.4% |
| Injury/Trauma | 65 | 9.8% | 67 | 9.4% | 64 | 7.9% | 189 | 10.1% |
| Malaria | 136 | 20.6% | 155 | 21.7% | 236 | 29.1% | 514 | 27.4% |
| Missing | 30 | 4.5% | 2 | 0.3% | 18 | 2.2% | 37 | 2.0% |
| Myalgia | 0 | 0.0% | 16 | 2.2% | 20 | 2.5% | 78 | 4.2% |
| Neuritis | 14 | 2.1% | 1 | 0.1% | 4 | 0.5% | 2 | 0.1% |
| Other | 64 | 9.7% | 95 | 13.3% | 81 | 10.0% | 218 | 11.6% |
| Pneumonia, Asthma, Pleurisy/LRTI | 5 | 0.8% | 9 | 1.3% | 29 | 3.6% | 35 | 1.9% |
| PUD | 25 | 3.8% | 19 | 2.7% | 10 | 1.2% | 21 | 1.1% |
| Sexual Dysfunction | 2 | 0.3% | 0 | 0.0% | | 0.0% | | 0.0% |
| STI | 14 | 2.1% | 23 | 3.2% | 24 | 3.0% | 43 | 2.3% |
| URTI | 136 | 20.6% | 190 | 26.6% | 224 | 27.6% | 516 | 27.5% |
| UTI | 46 | 7.0% | 18 | 2.5% | 20 | 2.5% | 45 | 2.4% |
| Total | 661 | | 715 | | 811 | | 1875 | |

correctly 25% of the time (Table 6). The proportion of correct treatment for diagnosis in the intervention group ranged from 25·0% (n = 3) for fungal infections to 100% (n = 10) for epilepsy.

The proportion of times PITC was offered in the intervention group was high for patients who did not know their HIV status (90·3%, n = 149), compared to those who already knew their HIV status (8·5%, n = 34), p<0·0001.

## Discussion

Women regularly access health services, either for their own health or for their children. In contrast, in most parts of the world, men do not access health services as frequently. This

**Table 6. Proportion of correct treatments for select diagnoses at Male Clinics.**

| | Number of correct treatments | Percent correct | Number of incorrect treatments | Percent incorrect |
|---|---|---|---|---|
| Malaria | 129 | 94.9 | 6 | 10.2 |
| Fungal infections | 3 | 25.0 | 9 | 75.0 |
| Hypertension | 7 | 36.8 | 12 | 63.2 |
| Diabetes | 2 | 50.0 | 2 | 50.0 |
| Epilepsy | 10 | 100.0 | 0 | 0.0 |
| Peptic ulcer disease | 14 | 73.7 | 5 | 26.3 |
| Missing | 31 | | | |

phenomenon is unrelated to a difference in need for services [12]. Most health programs in low-resource settings are designed for women and children, with few exceptions focusing on men as clients in their own right [21–22]. Based on findings from our early focus group discussions, cost of services, distance to the facility and associated cost of transport, as well as quality of services were reported as the main barriers to seeking care. We also found that men sometimes do not access health services because they associate care-seeking behavior with weakness, feel it is not worth missing a day's salary, or believe health centers to be places for women and children that are not catered to their direct needs. These findings are consistent with previous research [12, 23–25]. As a result, many men access health services primarily for curative care [26–27] and at an advanced stage of a health issue [16–17].

In response, we launched MCs to address men's concerns and to provide services tailored to their needs, with the goal of improving health-seeking behavior. While MCs could not directly respond to the issues reported during focus groups associated with distance to the facility or cost of transport, MCs did directly addressed the problems of service costs and wait times.

The number of total visits at MCs was significantly higher compared to regular clinics across all control groups. Additionally, among visits where the reason was known, more men in MCs accessed services for a check-up (20·2% of the time) compared to all control groups (1·2% of visits). The high rate of utilization of preventive care at MCs is important given the general trend of low uptake of health resources among men, particularly at early stages of a health issue. In MCs, the amount of check-ups might indicate that more men would likely access preventive health services in the context of MCs where all services are free—not just for pregnant women and children—and when the wait time is short.

The largest number of visits within the intervention group were to Sauri clinic. This could be explained by the fact that services at Sauri were already free in addition to the MCs. Also, unlike other facilities in the area, Sauri clinic is closest to the referral hospital, was recently constructed, and had a large staff with good management. Conversely, other clinics sometimes experienced commodities stock-outs and were smaller.

We also noted that half of MC visits were from men who had accessed services at MCs before. This is a desired positive outcome for health facilities that want to monitor progress of their patients in a context where most male patients fail to return for follow-ups, or want their patients to come in the early stages of any ailment. We could not compare this result to control groups because these data were not captured in general outpatient services.

Close to 10% of MC visits were from men who presented with an injury, which is about the same proportion as in all control groups. This is notable considering that MCs were only held once monthly, but the proportion was similar to control groups where patients could get services every day. This may indicate that these patients did not visit the regular clinic on the day of the injury, but waited for MC as a way of minimizing their costs and wait time for services.

Our analysis demonstrated that the quality of care, measured as the percentage of adequate treatment for six common diagnoses, was the lowest in MCs as compared to control groups, though correct prescribing was high across all groups. This could be due to myriad factors, including potential recording bias in the control facilities—only data from entries recording both diagnosis and treatment provided could be used for our analysis, which was a subset of all the entries in the registers that are often poorly filled out—potentially limited commodity availability in the MCs leading to unavailability of correct medicines and small sample size for certain conditions leading to disproportionately high rates of incorrect treatment (e.g. n = 19 total treatments for hypertension and n = 4 for diabetes).

During MC sessions, PITC was accepted at over 90% (n = 149) of visits where patients reported not knowing their HIV status. This is an added benefit of MCs, which could assist in

reaching the first goal of the UNAIDS 90-90-90 targets [28]. In control registers, data on which visits resulted in PITC are available but whether the patient had existing knowledge of his status was not reported.

Additionally, rates of STI screening were not significantly higher at MCs compared to controls, despite the private setting available. The positivity rate following STI screening at MCs was the same as in control groups, which is expected since the male population is comparable across groups.

Finally, some practices following the inception of MCs contradicted results from the focus group discussions. For example, when asked for common reasons men would go to a clinic, 3·2% (n = 4) said for check-ups. Conversely, 20·2% of visits to MCs were for check-ups, indicating that perhaps there is greater demand for preventive services than had been anticipated.

Future research could consider investigating whether the gender of a provider impacts uptake and types of services. If men prefer male providers for services similar to those provided in MCs, uptake might be hindered by the dearth of male service providers at lower tiers of the health workforce [29]. Further research could also explore the impact of no-cost services on male health-seeking behavior.

## Limitations

The main limitation of this study is the difference in information captured using the MC registers compared with control registers. We developed the MC registers to collect all information needed to demonstrate impact, but similar information was not always available in existing national registers which was the source of information for the control populations. As a result, "services offered," such as blood pressure or STI screenings, tests given, or revisits were not systematically recorded in the control data. Additionally, registers are commonly lost in clinics, and are typically not kept very long once they are full, which made collection of historic data challenging. For example, records from the "concurrent control at different clinics" group were only available for part of the needed duration—from July 2015-November 2015 instead of November 2014-November 2015. Second, for confidentiality purposes, we did not collect personal identifiers. As a result, only distinct visits were tracked because it was not possible to track individual patients. This prevented us from distinguishing whether a revisit at MC was for continuing treatment of an existing illness or for a new ailment. Third, the algorithm to determine quality of care only took into consideration diagnoses with specific treatment options and assumed accurate diagnosis. This may not accurately reflect the quality of care for all services provided. Additionally, comparison of MC data with historical controls can be confounded by other temporal changes that may have impacted health seeking behavior, but this was mitigated through the concurrent control clinic comparisons.

## Ethical considerations

For focus group discussions, consent of adult men (age 18 and above) was written. No consent was needed from men subsequently attending MCs as all services provided are standard services available and approved by the Kenyan Ministry of Health. However, for patient privacy, patient names or other identifiable information were never recorded.

The protocol was approved by the Columbia University Institutional Review Board under protocol numbers IRB-AAAM0256 and AAAO2750. The study was also approved in Kenya by the Office of the President/Ministry of Interior and Coordination of National Government under reference number CORR 3\R\5\att\200.

## Conclusion

This research shows that an intervention focusing specifically on men as health-seeking agents in their own right, instead of targeting men as part of a health package geared toward their female partners and children, can successfully lead to increased male attendance and return visits. Creating an enabling environment for men at health centers as this study did can therefore improve uptake of preventive care. The Male Clinics intervention successfully engaged men in regular check-ups despite the socially-constructed barriers of gender norms which paint care-seeking as feminine.

Whereas a woman's health-seeking behavior might be triggered by a key life event such as childbirth, men do not have an equivalent event that would initiate a defined interaction with the health system. Standardizing the concept of Male Clinics in low-income countries could be an effective way to (re)connect men with health services. In the many contexts where there is a financial barrier to providing free care to all men, Ministries of Health, particularly in countries where men rarely seek care, may consider designing a specific life-defining moment where every man should visit a health center and the care given would also be free, as it is during MCs. This could be when men reach a specific age or at the birth of every one of his children. This would provide the benefits of MCs in that it would increase the uptake of preventive care, and thus significantly limit the funding required.

## Supporting information

**S1 Appendix. Male Clinics consultation form.** This is the form that was filled out by Male Clinics staff at each patient visit.
(DOCX)

**S2 Appendix. Interview guide.** This is the interview guide used for focus group discussions.
(PDF)

**S3 Appendix. Focus group questionnaire.** This is the questionnaire used during focus group discussions.
(PDF)

**S4 Appendix. Consent form.** This is the consent form used to recruit patients for the study.
(PDF)

**S5 Appendix. Code book.** This is the code book describing all the codes from the dataset.
(DOCX)

**S1 Dataset. Male Clinics data_aggregation.** This is the entire dataset for the Male Clinics study.
(XLSX)

## Author Contributions

**Conceptualization:** Yanis Ben Amor.

**Data curation:** Justine Dowden, Ivy Mushamiri, Eric McFeely, Donald Apat, Jilian Sacks.

**Formal analysis:** Justine Dowden, Ivy Mushamiri, Eric McFeely, Jilian Sacks.

**Funding acquisition:** Yanis Ben Amor.

**Investigation:** Justine Dowden, Eric McFeely, Donald Apat, Yanis Ben Amor.

**Methodology:** Jilian Sacks, Yanis Ben Amor.

**Resources:** Yanis Ben Amor.

**Supervision:** Donald Apat, Yanis Ben Amor.

**Writing – original draft:** Justine Dowden.

**Writing – review & editing:** Ivy Mushamiri, Eric McFeely, Donald Apat, Jilian Sacks, Yanis Ben Amor.

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
