## [Decision Letter · Decision Letter 0]

7 Aug 2019

PONE-D-19-16765

The impact of “male clinics” on health-seeking behaviors of adult men in rural Kenya

PLOS ONE

Dear Dr Yanis Ben Amor ,

Thank you for submitting your manuscript to PLOS ONE. After careful consideration, we feel that it has merit but does not fully meet PLOS ONE’s publication criteria as it currently stands. Therefore, we invite you to submit a revised version of the manuscript that addresses the points raised during the review process.

We would appreciate receiving your revised manuscript by 30th Sept 2019. To enhance the reproducibility of your results, we recommend that if applicable you deposit your laboratory protocols in protocols.io, where a protocol can be assigned its own identifier (DOI) such that it can be cited independently in the future. For instructions see: http://journals.plos.org/plosone/s/submission-guidelines#loc-laboratory-protocols

We look forward to receiving your revised manuscript.

Kind regards,

Kwasi Torpey, MD PhD MPH

Academic Editor

PLOS ONE

2. Please provide additional details regarding participant consent of the focus group/interview part of your study. In the ethics statement in the Methods and online submission information, please ensure that you have specified (1) whether consent was informed and (2) what type you obtained (for instance, written or verbal, and if verbal, how it was documented and witnessed). If your study included minors, state whether you obtained consent from parents or guardians. If the need for consent was waived by the ethics committee, please include this information

3. Please include a copy of the interview guide used in the study, in both the original language and English, as Supporting Information, or include a citation if it has been published previously.

Reviewers' comments:

Reviewer's Responses to Questions

**Comments to the Author**

1. Is the manuscript technically sound, and do the data support the conclusions?

Reviewer #1: Partly

Reviewer #2: Yes

2. Has the statistical analysis been performed appropriately and rigorously? 

Reviewer #1: Yes

Reviewer #2: Yes

3. Have the authors made all data underlying the findings in their manuscript fully available?

Reviewer #1: Yes

Reviewer #2: Yes

4. Is the manuscript presented in an intelligible fashion and written in standard English?

Reviewer #1: Yes

Reviewer #2: Yes

5. Review Comments to the Author

Reviewer #1: This is an important study as most health services globally tend not to focus on men and worse so in low resource settings yet men need to be empowered and enabled to use services.

The design is sound and the literature reviewed shows the gaps that exit in in this discipline.

The statistical analysis that was conducted was appropriate and answered what the researcher set out to do.

Lines 231-236 and table 5 : Can the researchers clarify their definition of RTI. Pneumonia and pleurisy are these not RTIs.

Table 5: UTIs and STIs. Can the researchers clarify how these were differentiated and confirmed ( laboratory or syndromic).

Line 248: PITC was the second most common test done. In which facilities? Would be helpful to report comparison results from the intervention, comparison and historic data.

Quality of care section. Line 251

This is a very important section in this study. The researchers can enrich the paper by elaborating this section to show if there were statistical differences on this outcome for the intervention and comparison sites.

Service utilization:

Were the opening hours the same in the intervention and control sites? Can this be specified.

Conclusion section Line 356

Lines 364-367: The recommendations made in this section are not backed by findings from the study. Researchers need to modify this section to be more explicit in line with the study findings.

Reviewer #2: The manuscript is well written and it fills an important gap in the literature. The multi-disease approach to men’s health is an important area to further explore as health services in some African countries tend to be siloed due to donor priorities. The paper highlights the benefit of a general health approach (rather than a disease-based approach) to engage men in health services.

Specific comments/suggestions

1. Table 5 seems to lump HIV diagnosis with STIs. Is it possible to separate them? This would be helpful for those in the HIV field. Indeed, engaging men in HIV services has been deemed as a blind spot in the HIV response. It would be helpful if more details on HIV diagnosis are included in the paper.

2. In the section of reproductive health, I am not sure if it makes sense to compare results from the treatment group to “all” control groups, especially since I am assuming this would include the historical control group (line 244-245). It may be better to compare to separate control groups.

3. It is well established that men’s health seeking behaviors are influenced by masculinity norms. This comes out tangentially in the paper in lines 271-273. The paper highlights cost of services, distance to the facility, cost of transport, and quality of services as the main barriers. While this may have been the main barriers cited by men, it is important not to discount the gender norms issues cited in line 271-273. I suggest finding a way to make sure those are not lost in the manuscript.

4. Another important aspect in the manuscript is that the intervention was successful in engaging men in regular check ups (compared to controls). This also needs to be highlighted in the discussion and conclusion.

5. For circumcision services, it may be helpful to indicate if the sites are high volume or low volume sites and provide the range of clients. Typically, in high volume sites hundreds of boys/men may access VMMC services. That clarification is important to understand the context

6. The age range of the men participants is important. Also, if you can provide any short table with socio-demographic characteristics that would be a bonus to help the reader understand who are the men who accessed MC.

6. PLOS authors have the option to publish the peer review history of their article (what does this mean?). If published, this will include your full peer review and any attached files.

Reviewer #1: No

Reviewer #2: Yes: Maria A. Carrasco

---

## [Author Response · Author response to Decision Letter 0]

30 Sep 2019

Re: Response to reviewers for the submission titled, “The impact of ‘male clinics’ on health-seeking behaviors of adult men in rural Kenya”

September 30, 2019

To the Reviewers and Editors,

Thank you for your careful review of our submission titled, “The impact of ‘male clinics’ on health-seeking behaviors of adult men in rural Kenya.” Please find our point-by-point responses below.

JOURNAL REQUIREMENTS

Our manuscript meets PLOS ONE’s style requirements, including those for file naming. 

2. Please provide additional details regarding participant consent of the focus group/interview part of your study. In the ethics statement in the Methods and online submission information, please ensure that you have specified (1) whether consent was informed and (2) what type you obtained (for instance, written or verbal, and if verbal, how it was documented and witnessed). If your study included minors, state whether you obtained consent from parents or guardians. If the need for consent was waived by the ethics committee, please include this information

For the focus groups discussions, consent of adult men (age 18 and above) was written. This was added in the Ethical Considerations section (line 366).

3. Please include a copy of the interview guide used in the study, in both the original language and English, as Supporting Information, or include a citation if it has been published previously.

A copy of the interview guide (S2 Appendix Interview Guide), questionnaires of the focus groups (S3 Appendix Focus Group Questionnaire) and Consent form (S4 Appendix Consent Form) have been included as supplementary documents. 

We have uploaded the minimal anonymized data set (S5 Appendix Male Clinics data and S6 Appendix Code Book)

Consent for publication of raw data was not obtained but dataset is fully anonymous in a manner that can easily be verified by any user of the dataset. Publication of the dataset clearly and obviously presents minimal risk to confidentiality of study participants. It was not possible to obtain consent from participants because the quantitative data was not collected from individuals but from clinic registers and the unit of analysis is a clinic visit not a patient.

REVIEWER 1:

Lines 231-236 and table 5: Can the researchers clarify their definition of RTI. Pneumonia and pleurisy are these not RTIs.

We thank the reviewer for spotting this. Pneumonia and asthma are Lower Respiratory Tract Infections (LRTI). To prevent confusion, we removed in tables 1 and 5 the mention of RTI and specified URTI and LRTI, where appropriate. 

Table 5: UTIs and STIs. Can the researchers clarify how these were differentiated and confirmed (laboratory or syndromic).

The diagnosis of UTIs and STIs was syndromic, except specifically for syphilis where the Venereal Disease Research Laboratory (VDRL) test was used. 

Line 248: PITC was the second most common test done. In which facilities? Would be helpful to report comparison results from the intervention, comparison and historic data.

PITC was the second most common test offered in intervention clinics across the board except for Ramula where it was the most common. Tests offered were only recorded in the intervention group so we cannot compare across groups. We thus did the assessment in the intervention group only and found that there was no statistically significant difference in PITC offering by intervention clinic (p= 0.1152). A clarification on this point has been added to the “Reproductive health services” section (lines 259-262).

Quality of care section. Line 251

This is a very important section in this study. The researchers can enrich the paper by elaborating this section to show if there were statistical differences on this outcome for the intervention and comparison sites.

We did not disaggregate the quality of care outcome by control group because there was such a small difference amongst them. Instead we conducted the analysis by combining all control groups. More detail has been added to this effect in the “Quality of care” section, lines 267-269. For the reviewers’ reference, the proportion of correct treatments by diagnosis was 90.91% in the “concurrent control at different clinics” group, 93.3% in the historic control group, and 90.94% in the “concurrent control at intervention clinics” group.

Service utilization:

Were the opening hours the same in the intervention and control sites? Can this be specified.

Yes, the hours were the same. This has been added to the “Evaluation design” portion on line 150.

Conclusion section Line 356

Lines 364-367: The recommendations made in this section are not backed by findings from the study. Researchers need to modify this section to be more explicit in line with the study findings.

We thank the reviewer for this suggestion. We have rewritten this section to make it more in line with our findings about engaging men in preventive care. This is reflected in lines 382-391.

REVIEWER 2:

The manuscript is well written and it fills an important gap in the literature. The multi-disease approach to men’s health is an important area to further explore as health services in some African countries tend to be siloed due to donor priorities. The paper highlights the benefit of a general health approach (rather than a disease-based approach) to engage men in health services.

Thank you for your review and this comment. The reviewer’s last sentence here has been added to the introduction, lines 86-87.

Specific comments/suggestions

1. Table 5 seems to lump HIV diagnosis with STIs. Is it possible to separate them? This would be helpful for those in the HIV field. Indeed, engaging men in HIV services has been deemed as a blind spot in the HIV response. It would be helpful if more details on HIV diagnosis are included in the paper.

Thank you for this suggestion. Table 5 shows positive diagnoses for each condition, as opposed to the number of patients tested for the condition. For example, there were 11 patients diagnosed with diabetes in the intervention group. The number of patients tested for diabetes would be much higher. We did not record a patient’s positive or negative HIV status, rather we tracked whether a patient was aware of his HIV status. As a result, in Table 5, STI does not include HIV. Therefore, we are unable to disaggregate STI from HIV in the results or to report more thoroughly on HIV diagnoses of MC participants. However, we did record whether PITC was administered when a patient was unaware of his status, and that result is reported lines 279-281.

2. In the section of reproductive health, I am not sure if it makes sense to compare results from the treatment group to “all” control groups, especially since I am assuming this would include the historical control group (line 244-245). It may be better to compare to separate control groups.

We have done an assessment of STI diagnosis by individual groups (as opposed to combined control groups). We found no statistically significant difference in STI diagnosis by intervention or control group (p = 0.5043). The proportion of visits resulting in an STI diagnosis by group was added to the "Reproductive health services" section, lines 252-255. 

3. It is well established that men’s health seeking behaviors are influenced by masculinity norms. This comes out tangentially in the paper in lines 271-273. The paper highlights cost of services, distance to the facility, cost of transport, and quality of services as the main barriers. While this may have been the main barriers cited by men, it is important not to discount the gender norms issues cited in line 271-273. I suggest finding a way to make sure those are not lost in the manuscript.

We thank you for this suggestion. Further mention of this has been added to the conclusion, lines 379-381. 

4. Another important aspect in the manuscript is that the intervention was successful in engaging men in regular check ups (compared to controls). This also needs to be highlighted in the discussion and conclusion.

This has been highlighted in the discussion, lines 298-302 and conclusion, lines 379-381.

5. For circumcision services, it may be helpful to indicate if the sites are high volume or low volume sites and provide the range of clients. Typically, in high volume sites hundreds of boys/men may access VMMC services. That clarification is important to understand the context

Information about how VMMC played a role in the context of male clinics has been added to the Study area section, lines 102-104. 

6. The age range of the men participants is important. Also, if you can provide any short table with socio-demographic characteristics that would be a bonus to help the reader understand who are the men who accessed MC.

Because age was the only demographic variable measured, we did not add a table. We added a more detailed description of men who accessed MC in the “demographics” section, lines 217-219.

---

## [Editor Report · Decision Letter 1]

22 Oct 2019

The impact of “male clinics” on health-seeking behaviors of adult men in rural Kenya

PONE-D-19-16765R1

Dear Dr. Yanis Ben Amor,

We are pleased to inform you that your manuscript has been judged scientifically suitable for publication and will be formally accepted for publication once it complies with all outstanding technical requirements.

With kind regards,

Kwasi Torpey, MD PhD MPH

Academic Editor

PLOS ONE
---

## [Editor Report · Acceptance letter]

7 Nov 2019

PONE-D-19-16765R1 

The impact of “male clinics” on health-seeking behaviors of adult men in rural Kenya 

Dear Dr. Ben Amor:

I am pleased to inform you that your manuscript has been deemed suitable for publication in PLOS ONE. Congratulations! Your manuscript is now with our production department. 

With kind regards,

on behalf of

Professor Kwasi Torpey 

Academic Editor

PLOS ONE